# Fertility-Sparing Surgery in Gynecologic Cancer: A Systematic Review

**DOI:** 10.3390/cancers13051008

**Published:** 2021-02-28

**Authors:** Teska Schuurman, Sanne Zilver, Sanne Samuels, Winnie Schats, Frédéric Amant, Nienke van Trommel, Christianne Lok

**Affiliations:** 1Department of Gynecologic Oncology, Netherlands Cancer Institute, Antoni van Leeuwenhoek, 1066 CX Amsterdam, The Netherlands; t.schuurman@nki.nl (T.S.); s.samuels@nki.nl (S.S.); f.amant@nki.nl (F.A.); n.v.trommel@nki.nl (N.v.T.); 2Department of Gynecology, Amsterdam University Medical Center, 1105 AZ Amsterdam, The Netherlands; s.j.zilver@amsterdamumc.nl; 3Department of Scientific Information Service, Netherlands Cancer Institute, Antoni van Leeuwenhoek, 1066 CX Amsterdam, The Netherlands; w.schats@nki.nl; 4Department of Oncology, Catholic University of Leuven, 3000 Leuven, Belgium

**Keywords:** fertility-sparing surgery, conization, trachelectomy, hormonal therapy, cervical cancer, ovarian cancer, endometrial cancer, adolescents and young adults

## Abstract

**Simple Summary:**

In Adolescents and Young Adults (AYAs), fertility is an important factor for good quality of life. In the case of cancer of the female reproductive tract, treatment can impair fertility and therefore, AYAs may face the life-changing decision whether or not to undergo conservative, fertility-sparing cancer treatment. Solid evidence on the safety as well as reproductive outcomes of these treatments is necessary to provide patients the information they need to make a well-informed choice. This systematic review aims to provide an overview of the best evidence available on both oncological and reproductive outcome after various fertility-sparing surgical options in cervical, ovarian, and endometrial cancer.

**Abstract:**

Fertility-sparing surgery (FSS) is increasingly being offered to women with a gynecological malignancy who wish to preserve fertility. In this systematic review, we evaluate the best evidence currently available on oncological and reproductive outcome after FSS for early stage cervical cancer, epithelial ovarian cancer, and endometrial cancer. An extensive literature search was conducted using the electronic databases Medline (OVID), Embase, and Cochrane Library to identify eligible studies published up to December 2020. In total, 153 studies were included with 7544, 3944, and 1229 patients who underwent FSS for cervical, ovarian, and endometrial cancer, respectively. We assessed the different FSS techniques that are available to preserve fertility, i.e., omitting removal of the uterine body and preserving at least one ovary. Overall, recurrence rates after FSS are reassuring and therefore, these conservative procedures seem oncologically safe in the current selection of patients with low-stage and low-grade disease. However, generalized conclusions should be made with caution due to the methodology of available studies, i.e., mostly retrospective cohort studies with a heterogeneous patient population, inducing selection bias. Moreover, about half of patients do not pursue pregnancy despite FSS and the reasons for these decisions have not yet been well studied. International collaboration will facilitate the collection of solid evidence on FSS and the related decision-making process to optimize patient selection and counseling.

## 1. Introduction

Each year, over 1,300,000 women are diagnosed with a gynecologic malignancy worldwide. Nearly 15% of these women are between 15 and 39 years of age [1]. This age group is often referred to with the term AYAs (Adolescents and Young Adults), although cut-offs for both minimum and maximum age of AYAs vary between studies and countries. Nowadays, 80% of AYA cancer patients survive their disease because of the improved early detection and advancements in cancer treatment of many cancer types [2]. As a result, the focus of oncologic treatment has expanded from survival only towards quality of life after surviving cancer [3]. Preservation of fertility is an important factor to achieve good quality of life in AYAs [4]. However, fertility can be impaired by surgery or the gonadotoxic effects of chemotherapy and radiotherapy. Fertility is highly at risk in the case of treatment of malignancies of the female genital tract, especially cervical, ovarian, and endometrial cancer. Standard treatment for these cancer types often includes hysterectomy and bilateral salpingo-oophorectomy and, depending on stage, (adjuvant) therapy in the form of pelvic radiation or chemotherapy. Fertility-sparing surgery (FSS), in which ovaries, uterus, and sometimes cervix are (partially) preserved, is being offered in selected cases, e.g., in women with early stage malignant cervical, ovarian, and endometrial tumors.

### 1.1. Cervical Cancer

Of all cervical cancers, 20% is diagnosed in AYAs, making cervical cancer the most common gynecologic malignancy and the second cause of cancer-related death in these young women [1]. Fortunately, incidence and mortality are declining in some areas of the world due to population screening and Human Papilloma Virus (HPV) vaccination. However, more than 110,000 AYAs are diagnosed with cervical cancer and over 31,000 AYAs still die of the disease each year worldwide [1]. (Radical) hysterectomy with or without pelvic lymphadenectomy is considered standard treatment for early stage cervical cancer (FIGO 2018 IA1-IB2) in women who do not want to have children anymore. In women with a strong desire to preserve fertility, FSS options include conization or simple trachelectomy (cone or barrel-shaped excision of the cervix without surgery of the parametrium), or (vaginal or abdominal) radical trachelectomy (removal of the cervix, parametrium, and upper vaginal cuff), leaving the uterine body intact. These fertility-sparing procedures are increasingly being offered, as more and more studies suggest acceptable oncological outcomes comparable to radical hysterectomy [5]. However, the pregnancy rates especially after abdominal radical trachelectomy (ART) are disappointing [6,7] The number of live births after ART in stage IB2 is only 9% [8]. The pregnancy rate after neoadjuvant chemotherapy (NACT) followed by conservative surgery is promising, but oncological safety is still unclear [6,9,10].

### 1.2. Ovarian Cancer

Ovarian cancer is the 4th most commonly diagnosed cancer in AYAs [1]. Epithelial ovarian cancer is the most common type, although non-epithelial ovarian cancer occurs more often in young women than in women over 40 years of age [2]. The incidence of ovarian cancer in AYAs represents 13% of all new diagnoses annually. The implication is that approximately 38,500 young women are diagnosed with ovarian cancer and that 10,000 of these women die from the consequences of the disease each year worldwide [1].

The standard management of clinical early stage epithelial ovarian cancer is surgical staging, which includes hysterectomy, bilateral salpingo-oophorectomy, omentectomy, peritoneal washings, and biopsies with or without pelvic and para-aortic lymph node sampling. Depending on final stage and histology, platinum-based adjuvant chemotherapy can be proposed. During FSS, removal of the contralateral ovary and uterus are omitted [11]. This conservative treatment is considered in AYAs with a strong desire to preserve fertility and with limited disease and no visible abnormalities during surgery. However, only observational, retrospective series comparing FSS and non-FSS in selected patients are available and controversy about women with high-risk prognostic factors remains [12,13].

### 1.3. Endometrial Cancer

Although only 4% of endometrial cancers occur in AYAs, this is still a global incidence of approximately 15,000 newly diagnosed women each year, of which 1600 young women die of the disease [1].

In endometrioid endometrial cancer, standard management involves total hysterectomy and bilateral salpingo-oophorectomy, leading to very high cure rates of 93% in low-risk disease [2]. The fertility-sparing alternative treatment includes hysteroscopic resection and/or curettage in combination with hormonal therapy with progestin. Complete remission rates of this fertility-sparing approach of 50–75% have been reported, demonstrating the clear concession to the high effectiveness of the standard treatment by hysterectomy [14,15,16]. Strict follow-up with hysteroscopic evaluation and endometrial sampling is advised.

The level of evidence on oncological safety and chance of successful pregnancy after FSS in patients with a gynecological malignancy is low, because it is mainly based on retrospective case series including small numbers of patients and events. Moreover, follow-up is often short and incidence of pregnancy and pregnancy outcome are incompletely reported. This complicates adequate counseling of AYAs with gynecological cancer and the wish to preserve fertility and consequently, hampers these women to make the life-changing choice they are facing.

In this review, we evaluate the best evidence currently available on oncological and reproductive outcome after FSS for early stage cervical cancer, epithelial ovarian cancer, and endometrial cancer.

## 2. Materials and Methods

### 2.1. Search Strategy

We performed a systematic review in accordance with the PRISMA guidelines. This review was registered in the International Prospective Register of Systematic Reviews (PROSPERO, number CRD42020173523). A literature search was conducted using the electronic bibliographic databases Medline (OVID), Embase, and the Cochrane Library to identify eligible studies published between January 2000 and March 2020 in English or Dutch language. We searched for keywords and equivalent words in the title/abstract and translated the search terms according to the standards of each database. Keywords for uterine cervical neoplasms, ovarian neoplasms, and endometrial neoplasm were combined with terms for fertility-sparing treatments in general and trachelectomy specifically. The search string is presented in Appendix A. Reference lists of the included studies and retrieved review articles were searched to identify relevant articles not found in the initial search. Searches were re-run before the final analysis on the 3rd of December 2020.

### 2.2. Study Selection

The articles retrieved during the searches were screened for relevance on title/abstract and subsequently full text by two authors independently. Discrepancies were resolved by consensus after assessment by a third author. Rayyan, a systematic review web application, was used as a tool to screen the articles and to ensure the authors were blinded to each other’s decisions.

Included articles needed to specify oncological and/or reproductive outcomes after FSS, i.e., conization or (vaginal or abdominal) trachelectomy in cervical cancer, surgical staging with preservation of uterus and at least one ovary in epithelial ovarian cancer, and uterine preservation combined with hormonal therapy in endometrial cancer. Data published regarding fertility-sparing cancer treatment during pregnancy, in childhood cancer, or in non-epithelial ovarian cancer were excluded from our analysis.

The following studies were also excluded: (1) review articles without any new patient data, (2) case reports or small case series with less than 20 patients, (3) letters to editors, commentaries, or (conference) abstracts. Of the articles with duplicate patient information and articles updating prior published series, the articles with the most recent and complete data were included.

### 2.3. Data Extraction

A pre-defined data extraction form was used by three authors independently to collect and record the data of the selected studies. For each study, data regarding study characteristics (first author, year of publication, country), study design (type of study, data source, inclusion period, sample size), participants and tumor characteristics (age, tumor histology, FIGO stage), and intervention and outcome details were extracted.

### 2.4. Quality Assessment

The quality of the included articles was assessed using the Newcastle–Ottawa scale (NOS), which is suitable for assessing the quality of a non-randomized cohort and case–control studies [17]. This scale contains 8 items within 3 domains (selection, comparability, and exposure or outcomes) and the total maximum score is 9. A study receiving a score of 5 or more was considered as high quality and included in the final analysis. Three authors were involved in the quality assessment. The score of all initially included studies is presented in Appendix A.

### 2.5. Data Synthesis and Analysis

Descriptive statistics were used to analyze the frequencies, mean and median values of demographic data, and oncological and reproductive outcomes. Pearson’s chi-square test or Fisher’s exact test was used to compare the oncological outcome between subgroups. *p* < 0.05 was considered statistically significant.

## 3. Results

### 3.1. Study Characteristics

Of the 5598 screened articles, 153 met the predefined inclusion criteria and were assessed for their quality (Figure 1) [8,14,18,19,20,21,22,23,24,25,26,27,28,29,30,31,32,33,34,35,36,37,38,39,40,41,42,43,44,45,46,47,48,49,50,51,52,53,54,55,56,57,58,59,60,61,62,63,64,65,66,67,68,69,70,71,72,73,74,75,76,77,78,79,80,81,82,83,84,85,86,87,88,89,90,91,92,93,94,95,96,97,98,99,100,101,102,103,104,105,106,107,108,109,110,111,112,113,114,115,116,117,118,119,120,121,122,123,124,125,126,127,128,129,130,131,132,133,134,135,136,137,138,139,140,141,142,143,144,145,146,147,148,149,150,151,152,153,154,155,156,157,158,159,160,161,162,163,164,165,166,167,168]. A total of 16 studies (5 cervical, 3 ovarian, and 8 endometrial cancer) were excluded for final analysis due to a NOS score of 4 or less (Appendix A). All but 2 included studies were cohort studies, of which 16 had a prospective and 119 a retrospective design (of which 8 analyzed prospectively collected data). There was one matched case–control study on cervical cancer and one randomized controlled trial on endometrial cancer. Patients were recruited from medical records (*n* = 129) or national registries (*n* = 8). The included studies and their characteristics and outcome measures are listed in Appendix A.

### 3.2. Patient Characteristics and Treatment Modalities

The total study population of this review consists of 7544 women with cervical cancer, 3944 with epithelial ovarian cancer, and 1229 with endometrial cancer. The median ages are 31.6, 29.0, and 32.6 years, respectively. The histology, stage, and treatment modalities per tumor type are shown in Table 1, Table 2 and Table 3.

In patients with cervical cancer, squamous cell carcinoma (SCC) was the most common subtype. The majority of patients were diagnosed with stage IB1 (FIGO 2009) disease. FSS consisted of simple or radical trachelectomy, conization or large loop excision of the transformation zone (LLETZ), with or without pelvic lymphadenectomy. In 12 studies, patients received NACT prior to FSS. The majority of patients underwent an LLETZ, conization, or simple trachelectomy (ST). In total, 3.9% of patients (307/7851) initially planned for FSS had positive lymph nodes, positive resection margins, or other reasons contraindicating FSS. These patients underwent non-FSS treatment. In 2018, the FIGO staging for cervical cancer was revised [169]. All but one study ([26], *n* = 32) in this review used the former FIGO 2009 classification.

The majority of patients with epithelial ovarian cancer had a low-grade tumor (76.7%), mucinous histology (45%), and was diagnosed with stage IA or IB (52.6%). FSS consisted of unilateral salpingo-oophorectomy and preservation of the uterus and contralateral ovary, with or without additional staging procedures. In 18 studies, the completeness of surgical staging was specified. Complete peritoneal staging including peritoneal washings and/or biopsies and omentectomy ranged from 41 to 100% with an average of 70% (776/1110 patients). Pelvic and/or para-aortic lymph node sampling was performed in 56% (617/1110 patients). Of the patients with known details about the surgical technique, 275 of 426 (64.6%) underwent a laparotomy and 151 of 426 (35.4%) a laparoscopy. In total, 1815 of 3586 patients (50.6%) received adjuvant chemotherapy.

Almost all patients with endometrial cancer were diagnosed with grade 1 (92.4%), endometrioid adenocarcinoma (90.3%), and stage IA (83%). In 14 studies, the type of FSS prior to the start of hormonal therapy was reported and consisted of hysteroscopic resection (HR) and/or dilatation and curettage (D&C). In all 23 included studies, patients received hormonal therapy with progestin, although type, dosage, and treatment scheme differed considerably between studies. The majority of patients were treated with medroxyprogesterone acetate (MPA) or megestrol acetate (MA). MPA was used in 14 studies, all with different dosages, ranging from 20 to 1500 mg/day. In 17 studies, MA was used, in most patients with a dosage of 160 mg/day (range 40–800 mg/day). Patients received a levonorgestrel intrauterine device (LNG-IUD, 20 or 52 mg) as a single treatment in 5 studies (*n* = 91) and it was used in combination with oral progestin in 6 studies (*n* = 45). In 2 studies, metformin was given in addition to oral progestin and in another 2 studies, progestin was combined with GnRH analogues. The median duration of hormonal therapy with oral progestin ranged from 4.2 to 10.9 months and with LNG-IUD from 9.1 to 25.8 months.

### 3.3. Oncological Outcome

In Table 4, the oncological outcome after FSS in each of the gynecological malignancies is presented. Not all studies reported both recurrence and death from disease.

In 29 studies on cervical cancer, involving 2432 patients, LLETZ, conization, or ST with or without pelvic lymphadenectomy was performed. In total, 28 (3.6%) patients had recurrent disease and 6 (0.8%) died of disease during a median follow-up of 53 months (range 9–131 months). More detailed information was available of 23 recurrences. Significantly more recurrences occurred in patients with FIGO stage IB1 (3.1%) and IB2 (5.6%) disease than in patients with FIGO stage IA1 (0.2%) and IA2 (0.7%) disease. SCC recurred in 0.7% of patients, adenocarcinomas (AC) in 1.1%, and adenosquamous cell carcinomas (ASCC) in 4%. The differences between these histological subtypes are statistically not significant (*p* = 0.159).

In 31 studies, including 2401 patients, a vaginal radical trachelectomy (VRT) was performed. Recurrent disease occurred in 82 (4.2%) patients and 34 (1.7%) patients died, after a median follow-up of 52 months (range 9–131 months). Unfortunately, several studies did not report on tumor size and LVSI, or combined stage IA and IB1 disease, which are factors associated with recurrent disease. In 7 studies (*n* = 468 patients) reporting on recurrence rates, details on initial tumor size were described. The overall recurrence rate was 6.2%, divided into 3.5% in patients with tumors <2 cm and 20.5% in ≥2 cm (*p* ≤ 0.001). None of the patients with tumors <2 cm received NACT, although 5 of 83 patients with a tumor size ≥2 cm received NACT. In 12 studies (*n* = 1134), detailed information on recurrence according to histological subtype was described. The overall recurrence rate was 4.1%. In patients with SCC or AC, it was 3.4% and 6.1%, respectively (*p* = 0.03). In only 5 studies (*n* = 656), detailed information on initial LVSI in patients who had recurrent disease was reported. LVSI was found in 31.3% of patients. The overall recurrence rate was 3.8%. In tumors with LVSI vs. tumors without LVSI, recurrence rate was 5.1% vs. 3.0%, respectively (*p* = 0.16).

The abdominal approach for radical trachelectomy is more radical in terms of parametrial and paracervical resection compared to the vaginal approach. This procedure was originally offered to patients with less favorable prognosis, e.g., tumors >2 cm and LVSI. ART can be performed by laparotomy or as minimal invasive (laparoscopic or robot-assisted) procedure. In 28 studies, 2177 patients underwent ART per laparotomy. Recurrent disease was diagnosed in 47 (3.1%) patients and 23 (1.5%) deaths were reported. Median follow-up was 47 months (range 12–120 months). In 8 studies, more detailed information on recurrences was described. Based on these studies (*n* = 863 patients), recurrences were seen in 2.0% of patients with SCC, in 4.1% with AC, and in 2.7% with ASCC. These differences were not significantly different. Only one study found significantly more recurrences in ASCC [56]. In 3 studies (*n* = 367), detailed information was described on recurrences (*n* = 11) and tumor size. In total, 11 patients with tumors ≥2 cm received NACT. Overall, in 3.0% of patients, a recurrence occurred of 2.3% in patients with tumors <2 cm and 3.6% with tumors ≥2 cm (*p* = 0.74).

In 10 studies, involving 534 patients, a minimal invasive approach for ART was used. Recurrence and death rates were reported in 9 studies, with a median follow-up of 44 months (range 10–98 months). In total, 15 (4.5%) recurrences occurred and 7 (1.5%) patients died of disease. In 4 studies, including 177 patients, more detailed information on recurrences (*n* = 14) was reported, which were significantly more often found in patients with ASCC (28.6%) than with AC (4.2%) or SCC (7.3%). Almost all recurrences (*n* = 12) occurred in patients with FIGO stage IB1 disease. In 2 studies with 106 patients, tumor size was specified. In 4.3% of tumors <2 cm, recurrences occurred, compared to 18.9% of tumors ≥2 cm (*p* = 0.03).

Some studies (*n* = 24, including 3363 vs. 16,405 patients) compared oncological outcome after FSS with non-FSS (Appendix A). The disease-free survival (DFS) and overall survival (OS) or disease-specific survival (DSS) were not significantly different in both groups. In 15 studies, with 830 patients undergoing FSS and 1030 patients undergoing non-FSS, recurrences occurred in 2.4% vs. 3.3% patients, respectively. In one study, a trend was seen towards a worse DSS in FSS (conization or radical trachelectomy, *n* = 125) versus non-FSS (*n* = 2592) in patients with FIGO stage IB1 tumors >2 cm (82.4% vs. 90.4%, *p* = 0.112) [99].

In 21 of 24 studies with epithelial ovarian cancer patients, recurrence rates were reported and another 21 studies described death rates, with a median follow-up of 66 and 67 months (range 38–143), respectively.

In 15 studies, data about disease stage in relapsing patients were provided (Figure 2). In total, 40 of 406 patients (9.9%) with stage IA/IB and 44 of 285 patients (15.4%) with stage IC had a recurrence (*p* = 0.027). Stage IC is divided into three prognostic categories (IC1, IC2, and IC3) since the revised FIGO staging of ovarian cancer in 2014 [170]. Only 8 studies used this latest FIGO staging showing that 12 of 100 patients (12%) with stage IC1/2 and 8 of 21 patients (38.1%) with stage IC3 relapsed. This difference in recurrence rate is statistically significant (*p* = 0.003). The reported number of patients undergoing FSS with stage II or higher is extremely low, but 4 of the reported 10 patients (40%) with stage ≥II (stage II *n* = 2, stage III *n* = 2) relapsed compared to 88 of 639 patients (13.8%) with stage I (*p* = 0.040).

Death rate was 6.8% (99/1461) in stage IA/B versus 8.5% (68/800) in stage IC (*p* = 0.134), 3.2% (4/120) in stage IC1/2 versus 9.5% (2/21) in stage IC3 (*p* = 0.209), and 7.2% (168/2154) in stage I versus 39.2% (250/387) in stage ≥II (*p* ≤ 0.001) (Figure 2).

In 11 studies, the grade of the carcinomas in patients who suffered from recurrences was described and complete data were reported. In 62 of 622 (9.1%) patients with grade 1 or 2 carcinoma and in 32 of 125 (25.6%) patients with grade 3 carcinoma (including clear cell histology), the cancer recurred (*p* ≤ 0.001). From data of 10 studies, grade 3 non-clear cell histology could be analyzed separately. These studies showed a recurrence rate of 55% (11 of 20 patients, *p* ≤ 0.001).

Death rate was 8.8% (219/2489) in grade 1–2 carcinomas. This is statistically significantly lower compared to grade 3 (including clear cell) carcinomas (213/711, 30.0%, *p* ≤ 0.001), but not to grade 3 non-clear cell carcinomas (5/26, 19.2%, *p* = 0.063). A high percentage (40%) of stage II or higher with worse prognosis was included in the group of grade 3 carcinomas, including clear cell histology, influencing this subgroup analysis. When limiting these calculations to stage I disease, death rate was 5.8% (115/1987) in grade 1–2 carcinomas, which is statistically significant for grade 3 (including clear cell) carcinomas (14.1%, 57/405, *p* ≤ 0.001) but not for grade 3 non-clear cell carcinomas (12.5%, 2/16, *p* = 0.239).

In total, 12 of 85 patients (14.1%) with clear cell carcinoma relapsed compared to 82 of 673 (12.2%) patients with non-clear cell carcinoma (*p* = 0.610). Death rates were also not statistically different (5/106, 4.7% vs. 55/933, 5.9%, respectively, *p* = 0.622).

Often the localization of the recurrence was not reported. In 67 patients with known location of the recurrence, an isolated ovarian tumor was found in 23 patients (34.3%) and other sites of recurrence with or without involvement of the remaining ovary in 44 patients (65.7%). Recurrences occurring in patients with initially low-risk tumors (stage IA, grade 1–2, or non-clear cell histology) were more frequently isolated (62%). There were no recurrences recorded involving the uterus.

Some studies (*n* = 13) compared oncological outcome after FSS with radical, non-FSS (Appendix A). The recurrence rate of all patients who underwent FSS was 15.7% (171/1084) compared to 18.5% (266/1436) after non-FSS (*p* ≤ 0.001). Death rates were also lower in FSS, with 14.7% (489/3318) after FSS and 21% (1841/8774) after radical surgery (*p* ≤ 0.001).

In 21 of 23 studies on endometrial cancer, the response rate was reported with a median follow-up of 53.8 months. Complete response (CR) occurred in 80.2% of patients (736/918). The median time to CR was 5.5 months (range 3–9 months, reported in 17/21 studies). The remaining patients (182/918, 19.8%) had persistent disease. In 13 of the 21 studies (*n* = 504), the response in these patients was further divided into partial response (*n* = 16, 3.2%), stable disease (*n* = 52, 10.3%), and progressive disease (*n* = 28, 5.6%).

In 21 studies, recurrence rates in patients with initial CR were described, which was 34.7% (297/855). The 5-year recurrence-free survival (RFS) ranged from 33 to 68% in 5 studies. Although death rate was reported in 12 studies, in only 1 study, deaths occurred (5/648, 0.8%).

The study by Chung et al. is the only study in which a new classification of endometrial cancer, based on molecular subtypes, was used [143]. This ProMisE classification distinguishes endometrial cancer with (1) mismatch repair deficiency (MMRd), (2) DNA polymerase epsilon (POLE) mutation, (3) wild-type p53 (p53wt), and (4) abnormal p53 (p53abn) [171]. Of the 57 patients included in this study, 9 (15.8%) had a tumor with MMRd, 2 (3.5%) had a tumor with POLE mutation, 45 (78.9%) tumors were p53wt, and 1 tumor (1.8%) had p53abn. The overall CR rate to hormonal therapy was 75.4%. Patients with MMRd tumors had a significantly lower complete or partial response rate than those with p53wt (44.4% vs. 82.2%, *p* = 0.018). There was no difference in recurrence rate after achieving CR between patients with MMRd and p53wt (25.0% vs. 43.2%, *p* = 0.629). The oncologic outcomes were not compared for POLE and p53abn subtypes because of the small number of patients.

Only one study, by Greenwald et al., compared hormonal therapy with primary surgery, using the Surveillance, Epidemiology, and End Results (SEER) database [149]. Cancer-specific death rates were 3.1% (5/161) and 0.7% (46/6178), respectively (*p* = 0.001). However, when comparing to a propensity score-matched cohort (1/161, 0.6%), the difference was not statistically significant (*p* = 0.099). Recurrence rates were not reported.

In 2 studies, patients with grade 2 or 3 carcinoma were included and data about grade in patients who had CR were provided. Of 45 patients with grade 2–3 carcinoma, 37 (82.2%) had CR compared to 531 of 659 patients (80.6%) with grade 1 carcinoma (*p* = 0.787). Of these 37 patients, 15 (40.5%) had a recurrence compared to 188 of 509 patients (36.9%) with grade 1 carcinoma and CR (*p* = 0.661).

In 3 studies, LNG-IUD as a single treatment was used. The CR rate with LNG-IUD was 84% (63/75) versus 78.9% (435/551) with oral progestin (*p* = 0.309). The recurrence rate in these patients with CR was 9.5% (6/63) with LNG-IUD compared to 45.7% (217/475) with oral progestin (*p* ≤ 0.001). There was no difference in treatment effect between 20 or 52 mg LNG-IUD, although numbers were small. Due to heterogeneity in treatment strategies and lack of available data, no adequate subgroup analysis could be performed on type or dosage of oral progestin therapy.

### 3.4. Reproductive Outcome

Table 4 shows the reproductive outcome after FSS per tumor type. Most studies did not report if patients had a wish to conceive and attempted pregnancy after they underwent FSS. Of the studies that did, 51% of patients with a gynecological malignancy had a pregnancy wish after FSS.

In patients with cervical cancer, 36–55% reported a pregnancy wish after FSS. A vaginal approach of FSS in cervical cancer is thought to have the best reproductive outcome, possibly due to less extensive resection of the cervix/parametrium. In patients who underwent LLETZ, conization, or ST, the pregnancy rate was 59%, with a live birth rate of 77%. After VRT, the pregnancy rate and live birth rate were 66% and 71%, respectively. The lowest pregnancy rate (45%) and live birth rate (58%) were found after ART by laparotomy, which is significantly lower than other types of FSS. In the minimal invasive approach, there was a pregnancy rate of 59%, with a live birth rate of 72%.

After the least radical FSS (LLETZ, conization, or ST), fetal loss was 15%, which is significantly lower than after VRT (21%), laparotomic ART (25%), and minimal invasive ART (24%). Furthermore, the risk of preterm delivery was also significantly lower and was 15%, 30%, 46%, and 55%, respectively.

In epithelial ovarian cancer, pregnancy wish after cancer treatment was reported in 6 studies and ranged from 25 to 65.4%, with an average of 44.2%. The live birth rate varied from 57.1 to 100%.

In 12 studies on endometrial cancer, 62.6% of patients with CR on hormonal therapy had a pregnancy wish (range 36.7–100%). Of the patients with CR, 36.9% became pregnant (reported in 17 studies).

## 4. Discussion

In this study, we summarized and analyzed the available literature on FSS in gynecological cancers known to date and performed a critical quality assessment, resulting in this overview of fertility-sparing options in three different types of gynecological malignancies.

Mostly, retrospective cohort studies were found describing a heterogeneous patient population. Next to that, information on recurrences and deaths are often missing or not reported per subgroup, which complicates comparison between studies and groups of patients. With its limitations in mind, some important conclusions can be drawn from our analysis.

One of the most important conclusions is that we need a structured, well-developed international database with predefined variables before we can optimally counsel our patients based on solid data.

A second conclusion is that many of the patients that undergo FSS to preserve fertility choose not to achieve a pregnancy after treatment. This decision is likely to be influenced by many factors, not all known before treatment. However, previous reports on fertility preservation in cancer patients undergoing ovarian tissue cryopreservation also showed that only a minority of women will use the offered possibility of achieving pregnancy this way [172]. Apparently, this is also true for gynecological cancer as only 44–63% of the patients in our analysis were shown to have a pregnancy wish after treatment. It is important to obtain more insight into the reasons women undergo FSS procedures, thereby deviating from standard treatment and sometimes accepting increased risk of recurrence or death, and not pursuing pregnancy even if their outcome is reassuring.

A third conclusion we can draw is that in general, the recurrence rate after FSS for gynecological cancer is small, sometimes even smaller than after standard treatment, but selection of patients with good prognosis is likely. Therefore, conclusions should be generalized with caution.

In this systematic review, we have found that in patients with early stage cervical cancer (majority FIGO 2009 stage IA-IB1), the chance for recurrence after FSS varies between 3.1 and 4.5% after the several evaluated treatment modalities and 0.8–1.7% died of their disease. Two groups showed worrying recurrence rates: women with carcinomas ≥2 cm treated with VRT (20.5%) or with minimal invasive ART (18.9%). Internationally, the chance for recurrence after non-FSS in early stage cervical cancer is reported to be 6–9% [173,174]. In some studies included in this review, which provided data on both FSS and non-FSS, recurrence and death rate in both groups are not reported to be different. The risk of recurrence in cervical carcinoma is mostly influenced by tumor size, nodal status, and tumor characteristics, such as deep stromal invasion [175]. In the studies included in our review, information on these prognostic factors is sporadically given. Since FSS may harbor an increased risk for recurrence due to less radical treatment, one can imagine that patients with poor prognostic factors are offered FSS less frequently. The reassuring outcomes we found regarding the risk of recurrence may reflect a good selection of patients with favorable prognostic factors rather than the oncological safety of FSS itself.

Our review shows that ART per laparotomy has the lowest pregnancy rate (45.0%) compared to all other treatment modalities with pregnancy rates around 60%. As expected, the radicality of cervical surgery is related to the risk of preterm delivery. As more radical surgery is performed in larger sized cervical tumors, treatment with NACT to reduce tumor size, enabling less radical surgery in larger tumors, may be beneficial in reducing the risk of prematurity. The safety and feasibility of this combined treatment for patients with FIGO 2018 IB2 cervical carcinoma is now studied in the prospective CONTESSA/NEOCON-F trial and results are awaited [10].

In ovarian cancer, the same phenomenon of favorable patient selection is seen. Prognostic factors include stage, histological subtype, and grade. Unfortunately, most studies are too small to allow reliable subgroup analysis. The majority of patients in our review had stage 1A or IB epithelial ovarian cancer with a recurrence rate of 9.9%. Intra-abdominal spill or ascites (stage IC) significantly increased the risk of recurrence to 15.4%. However, many studies did not use the new FIGO staging classification that further subdivides stage IC into stage IC1, IC2, and IC3. This discrimination is important as we found 38.1% recurrences in stage IC3 vs. 12% in stage IC1/2. As the number of available patients for analysis was minimal (*n* = 21), the controversy on whether it is safe to offer FSS to young women with stage IC continues. Given the tendency of ovarian cancer to recur transperitoneally, it is hypothesized that grade and stage, but not FSS, add to the risk of relapse.

The incidence of different histological subtypes in AYAs is different from older women. Non-epithelial histology occurs more often in young women. We did not include these malignancies in our review because of their different biological behavior and often more favorable prognosis. This is why there is a general acceptance to offer FSS in these women in cases of low-stage disease. Inclusion in the current analysis would have clouded our results. The majority of patients in our review had mucinous histology and numbers of patients with serous or clear cell histology were much lower. Nowadays, mucinous tumors are divided into expansile vs. infiltrative types. Only one small study discriminated between these types and found no difference in recurrence rate. Adherence to the new WHO classification system for ovarian cancer is necessary to learn whether FSS for all mucinous tumors is safe.

In epithelial ovarian cancer, grade is very important in estimating the chance of recurrence. FSS for high-grade tumors cannot be safely advised as we found 25.6% recurrences in grade 3 carcinomas (including clear cell histology) compared to 9.1% in grade 1–2. It is unlikely that the high recurrence rate in grade 3 carcinomas is only explained by inclusion of clear cell histology, as in our analysis, the recurrence rate in clear cell carcinoma was 14.1% versus 12.2% in non-clear cell carcinoma. Unfortunately, most studies do not report detailed characteristics of the patients that develop recurrences. If a recurrence can be curatively treated, survival is not compromised. This is often the case if the recurrence develops in the remaining ovary. We found 34.3% isolated recurrences in the contralateral ovary. In extra-ovarian recurrences, it is disputable whether the FSS caused the recurrence or that the metastatic spread had already occurred and would not have been prevented by the removal of the uterus and contralateral ovary. This is then merely a reflection of the biological behavior than the surgical technique, which is confirmed by our findings that isolated recurrences occurred more in low-risk subtypes and extra-ovarian recurrences more in high-risk subtypes (grade 3, stage IC, clear cell carcinoma). If staging is inadequate, stage III disease can be missed and lead to recurrent disease. In our group, 70% of patients underwent staging with at least omentectomy, peritoneal biopsies, and washings. In 56% of patients, lymph node sampling was performed. Adjuvant chemotherapy was started in almost 51% of patients. Therefore, it is not the FSS itself that is evaluated, but the fertility-sparing treatment as a whole. Despite these limitations, our review includes almost 4000 ovarian cancer patients undergoing FSS, and with the current selection of patients, the procedure seems oncologically safe in low-grade disease. Still, careful consideration remains important, because also after FSS in ovarian cancer, only 44.2% of the patients have a pregnancy wish and 29.5% achieve a pregnancy.

Early stage (endometrioid) endometrial cancer has an excellent prognosis after treatment with hysterectomy and bilateral salpingo-oophorectomy. Hormonal therapy is less effective compared to definitive surgery, although the response rate is 80% in this review. This rate is comparable to previous reviews of Gunderson et al. [15] and Gallos et al. [176], who reported a response rate of 74.6% and 76.2%, respectively. In most studies, patients with complex hyperplasia are included besides endometrial cancer, which will give a more optimistic response rate and therefore, differ from the current analysis. Although hormonal therapy can be an acceptable alternative in terms of initial response, recurrence rate is very high, reaching almost 35%. Therefore, it is important that patients pursue pregnancy soon after remission and that hysterectomy is performed after completion of childbearing [177]. There is no consensus regarding optimal treatment strategy, because different types of progestin, dosage, and administration are used. The chance of complete remission with LNG-IUD is not significantly different compared to oral agents. However, the risk of recurrence is higher after oral administration (45.7% versus 9.5% after LNG-IUD). A possible explanation for this difference is that in two of the 3 studies investigating LNG-IUD, a hysteroscopic resection was also performed, which was not standard in patients who started with oral treatment. In previous studies, this procedure increased the response rate and reduced the chance of recurrence [178]. Most studies are single arm studies which obstructs a proper comparison. It is unknown if a combination of LNG-IUD and oral agents further improves outcome. Metformin can be added to the conservative treatment of endometrial cancer. Women most likely to benefit from this strategy are women with obesity, polycystic ovary syndrome, and insulin resistance. It has been previously reported that obese patients with endometrioid endometrial cancer have less risk of cancer recurrence on metformin [179] and for patients with endometrial cancer, the use of metformin was associated with improved RFS and OS [180]. However, groups are heterogeneous (often also including hyperplasia) and the only available RCT does not show a benefit from the addition of metformin [167].

The treatment of endometrial cancer is rapidly changing as molecular markers are being integrated in the (adjuvant) treatment as they predict the prognosis of patients. In FSS, these markers are not evaluated yet, although they may predict response to conservative treatment. Patients with MMRd tumors may be less responsive than with POLE mutation. Thus far, numbers are too small to draw any conclusions.

We found a relatively high number of fetal losses (31.3%) after FSS for endometrial cancer, but the low number of studies describing obstetric outcome may influence this number. The live birth rate of 72% is comparable to other studies [181].

## 5. Conclusions

From the above, it becomes clear that counseling by experts is required to help patients making the life-changing decision whether or not to undergo FSS. However, it is difficult to become an expert, as the situation is rare. According to the WHO, a disease is rare if the number of affected people is less than 5 per 10,000. The care for patients with rare diseases is challenged by the small number of patients, lack of validated treatment options, scattering of patients across a country, and limited clinical expertise. Although cervical cancer, ovarian cancer, and endometrial cancer do not meet the official criteria of a rare disease, the care for AYAs with gynecological cancer and a wish to preserve fertility is a unique situation encountering the same problems described for rare diseases. Tailor-made medicine in oncological care is a rapidly growing approach where, based on malignancy-specific characteristics, individualized treatment is given. Patient-specific characteristics are equally important to consider for a specific treatment. AYAs have their unique social situations, personal beliefs, and preferences. The outcome of a process of shared decision-making is important for tailor-made care. Decisions on fertility are not only difficult for the patient, but also for the treating gynecologist. Therefore, management of these patients should involve a (experienced) multidisciplinary team. It is generally accepted that exposure of a medical specialist to a rare health problem or situation is related to knowledge and quality of care. Participation in clinical trials or registration studies is indispensable, as this is the only way to evaluate and improve quality of care. International collaboration will facilitate achieving larger patient numbers and development of guidelines.

## Figures and Tables

**Figure 1 cancers-13-01008-f001:**
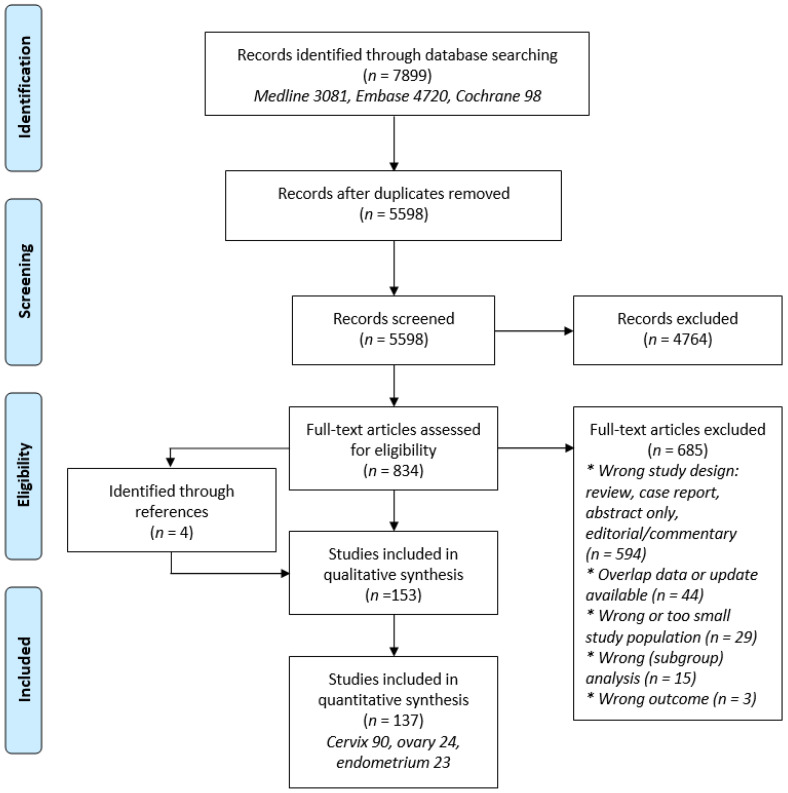
PRISMA flowchart summarizing the process for the identification of eligible articles.

**Figure 2 cancers-13-01008-f002:**
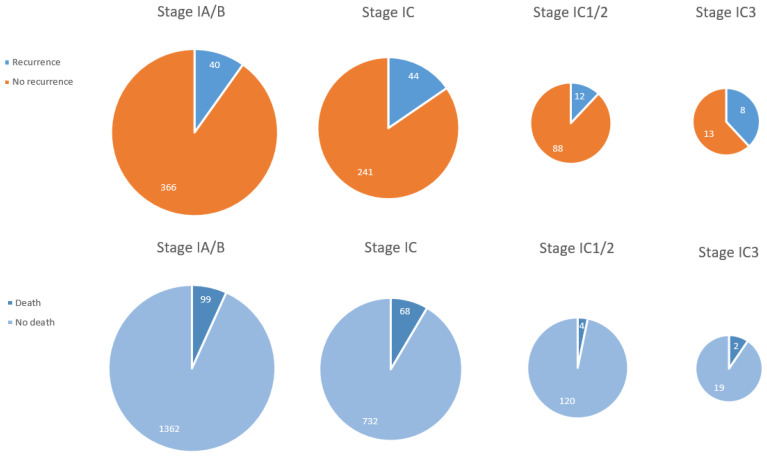
Recurrence rate and death rate in epithelial ovarian cancer per FIGO stage.

**Table 1 cancers-13-01008-t001:** Main characteristics of patients with cervical cancer undergoing FSS.

	LLETZ/CKC/ST	VRT	ART	
			Laparotomic	MIS
**Study/patient characteristics**				
*n* of studies	29	31	28	10
Total *n* of patients	2540	2477	2268	566
*n* of patients excluded #	108	76	91	32
Age (y), median (range)	32 (22–46)	31 (21–42)	32 (22–40)	31 (22–40)
**Tumor characteristics ^**	***n* of patients (%)**	***n* of patients (%)**	***n* of patients (%)**	***n* of patients (%)**
FIGO stage *				
IA1	1551 (60.4)	248 (9.0)	165 (7.1)	13 (2.4)
IA2	547 (21.3)	311 (11.3)	218 (9.4)	50 (9.3)
IA-IB	0	230 (8.4)	143 (6.2)	0
IB1	452 (17.6)	1933 (70.3)	1723 (74.6)	453 (84.0)
IB2	18 (0.7)	3 (0.1)	20 (0.9)	3 (0.6)
IB nos	0	0	17 (0.7)	19 (3.5)
IIA	2 (0.1)	26 (0.9)	25 (1.1)	1 (0.2)
Histological subtype				
SCC	2316 (82.2)	1906 (67.9)	1689 (72.4)	339 (63.2)
AC	467 (16.6)	770 (27.4)	518 (22.2)	169 (31.5)
ASCC	25 (0.9)	86 (3.1)	84 (3.6)	12 (2.2)
Other	8 (0.3)	47 (1.7)	43 (1.8)	16 (3.0)
LVSI~	150/784 (19.1)	586/1877 (31.2)	657/1628 (40.4)	59/341 (17.3)

FSS—fertility-sparing surgery; LLETZ—large loop excision of transformation zone; CKC—cold knife conization; ST—simple trachelectomy; VRT—vaginal radical trachelectomy; ART—abdominal radical trachelectomy; MIS—minimal invasive surgery (laparoscopic or robot-assisted); SCC—squamous cell carcinoma; AC—adenocarcinoma; ASCC—adenosquamous cell carcinoma; LVSI—lymph-vascular space invasion; nos—not otherwise specified; # No FSS due to positive lymph nodes/margins or other reasons contraindicating FSS; ^ In some studies, this also includes excluded patients or patients who underwent other types of FSS if data were not reported separately; therefore, the *n* of patients could exceed the total *n* of patients; * FIGO 2009 classification was used in all but one study, which used FIGO 2018 (*n* = 32); ~limited to publications with available data.

**Table 2 cancers-13-01008-t002:** Main characteristics and treatment modalities of patients with epithelial ovarian cancer undergoing FSS.

Patient Characteristics	
Total *n* of patients	3944
Age (y), median (range)	29 (23.5–36.5)
**Tumor characteristics**	***n* of patients (%)**
FIGO stage	
IA	912 (23.1)
IB	9 (0.2)
IA/IB	1154 (29.3)
IA/IC1	18 (0.5)
IC nos	808 (20.5)
IC1	220 (5.6)
IC2	35 (0.9)
IC3	25 (0.6)
I nos	9 (0.2)
≥II	646 (16.4)
na	108 (2.7)
Histological subtype	
mucinous	1775 (45.0)
endometrioid	1059 (26.9)
serous	760 (19.3)
clear cell	287 (7.3)
mixed	15 (0.4)
other/na	48 (1.2)
Grade	
1–2	3015 (76.4)
2–3	6 (0.2)
3 (+clear cell)	855 (21.7)
*3 non-clear cell*	*568 (14.4)*
na	68 (1.7)
**Treatment modalities ***	***n*/total (%)**
Type of FSS	
staging	
Complete #	776/1110 (69.9)
LND ^	617/1110 (55.6)
Surgical technique	
laparotomy	275/426 (64.6)
laparoscopy	151/426 (35.4)
Chemotherapy	
adjuvant	1815/3586 (50.6)

FSS—fertility-sparing surgery; LND—lymph node dissection; nos—not otherwise specified; na—not available; * limited to publications with available data; # unilateral salpingo-oophorectomy or cystectomy, cytology, biopsies, and omentectomy; ^ pelvic and/or para-aortic.

**Table 3 cancers-13-01008-t003:** Main characteristics and treatment modalities of patients with endometrial cancer undergoing FSS.

Patient Characteristics	
Total *n* of patients	1229
Age (y), median (range)	32.6 (31–36.1)
**Tumor characteristics**	***n* of patients (%)**
FIGO stage	
IA	1020 (83.0)
IB	6 (0.5)
IB-IIA	15 (1.2)
I nos	188 (15.3)
Histological subtype	
endometrioid	1110 (90.3)
adenosquamous	10 (0.8)
mucinous	1 (0.1)
mixed	8 (0.7)
adenocarcinoma nos	35 (2.8)
na	65 (5.3)
Grade	
1	1136 (92.4)
2	86 (7.0)
3	4 (0.3)
na	3 (0.2)
**Treatment modalities ***	***n* of patients (%)**
Type of FSS	
D&C	245 (19.9)
HR	155 (12.6)
HR + D&C	57 (4.6)
NR	772 (62.8)
Hormonal therapy	
MPA	458 (37.3)
MA	337 (27.4)
LNG-IUD	91 (7.4)
MPA + MA	7 (0.6)
MPA/MA + LNG-IUD	26 (2.1)
MPA/MA + metformin	57 (4.6)
MPA/MA nos	198 (16.1)
Progestin nos	51 (4.1)
Other ^	4 (0.3)

FSS—fertility-sparing surgery; HR—hysteroscopic resection; D&C—dilatation and curettage; MA—megestrol acetate; MPA—medroxy-progesterone acetate; LNG-IUD—levonorgestrel intrauterine device; nos—not otherwise specified; na—not available; NR—not reported; * Limited to publications with available data; ^ norethisterone acetate (*n* = 2) and hydroxyprogesterone caproate (*n* = 2).

**Table 4 cancers-13-01008-t004:** Oncological and reproductive outcome after FSS in patients with cervical, ovarian, and endometrial cancer.

	Cervical Cancer				Ovarian Cancer	Endometrial Cancer
	LLETZ/CKC/ST	VRT	ART			
			Laparotomic	MIS		
**Oncological outcome ***						
Recurrence	28/785 (3.6)	82/1973 (4.2)	47/1520 (3.1)	15/335 (4.5)	171/1084 (15.7)	297/855 (34.7)
Died of disease	6/785 (0.8)	34/1973 (1.7)	23/1520 (1.5)	7/479 (1.5)	489/3318 (14.7)	5/648 (0.8)
Complete response	–	–	–	–	–	736/918 (80.2)
Persistent disease	–	–	–	–	–	182/918 (19.8)
PR	–	–	–	–	–	16/504 (3.2)
SD	–	–	–	–	–	52/504 (10.3)
PrD	–	–	–	–	–	28/504 (5.6)
Follow-up in months,	53 (9–137)	52 (9–131)	47 (12–120)	44 (10–98)	66 (38–143)	56 (17–92)
median (range)						
**Fertility outcome ***						
Pregnancy wish ^	148/305 (48.5)	599/1089 (55.0)	510/1062 (48.0)	71/199 (35.7)	148/335 (44.2)	279/446 (62.6)
Pregnancies ^	241/612 (39.4)	707/1539 (45.9)	353/1635 (21.6)	81/344 (23.5)	280/750 (37.3)	256/505 (50.7)
Pregnant patients ^	196/665 (29.5)	512/1392 (36.8)	255/1427 (17.9)	58/317 (18.3)	207/701 (29.5)	261/708 (36.9)
Pregnancy rate ^#^	82/138 (59.4)	361/546 (66.1)	212/471 (45.0)	42/71 (59.2)	110/148 (74.3)	161/241 (66.8)
Fetal loss ~	36/234 (15.4)	147/686 (21.4)	101/407 (24.8)	19/81 (23.5)	17/191 (8.9)	67/214 (31.3)
Termination ~	7/234 (3.0)	26/686 (3.8)	20/407 (4.9)	0/81 (0)	3/191 (1.6)	0/214 (0)
Live birth ~	181/234 (77.4)	484/686 (70.6)	236/407 (58.0)	58/81 (71.6)	236/264 (89.4)	172/239 (72.0)
Term delivery ^$^	111/181 (61.3)	262/484 (54.1)	80/236 (33.9)	26/58 (44.8)	161/164 (98.2)	82/93 (88.2)
Preterm delivery ^$^	27/181 (14.9)	146/484 (30.2)	109/236 (46.2)	32/58 (55.2)	3/164 (1.8)	9/93 (9.7)
Not specified	43	76	47	0	72	79
Pregnancy ongoing	10	20	25	4	7	2

*n*/total (%); ^ total patients; # total patients with pregnancy wish who succeeded; ~ total pregnancies; $ total live births; LLETZ—large loop excision of the transformation zone; CKC—cold knife conization; ST—simple trachelectomy; VRT—vaginal radical trachelectomy; ART—abdominal radical trachelectomy; MIS—minimal invasive surgery (laparoscopic or robot-assisted); PR—partial response; SD—stable disease; PrD—progressive disease; * Limited to publications with available data.

## Data Availability

Not applicable.

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
