# Peer review of "Fertility-Sparing Surgery in Gynecologic Cancer: A Systematic Review"

_cancers, 2021, doi:10.3390/cancers13051008_

Round 1

Reviewer 1 Report

This is an in-depth systematic review of literature pertaining to fertility sparing surgery and the oncologic and reproductive outcomes in 3 gynecological cancers. There is high level of detail and analysis here, covering various aspects such as tumour grade, cancer stage, molecular subtypes, surgical options, cancer recurrence rate, death rate, pregnancy outcomes and various other factors. The reviewed articles are grouped together for numerical/descriptive analysis, while some studies are examined individually. This review is well-presented and written, and despite the many variables and outcomes that are listed, there is a logical flow to the report. The limitations such as selection bias, lack of data, small sample sizes are clearly explained to avoid drawing conclusions that may not be valid.

Overall, this is an impressive work and evaluation of FSS using the systemic review approach.  

Table 4 ties a lot of the data together - but there seems to be some minor discrepancies: for instance

82 / 1973 (4.3)

and 

34 / 1973 (1.8)

Should the percentages here be 4.2 and 1.7, respectively?  

Line 288, the median follow-up time of 31 months seems different to what is in Table 4, which is 44 months, listed under MIS. This may be a misreading of this line?  

In 10 studies, involving 534 patients, a minimal invasive approach for ART was used. 288
The median follow-up time was 31 months (range 22-40 months).

There are various mentions of "life birth" but I am more familiar with "live birth" and wondering if this is a translation issue?

On line 77, there is a formatting issue: 

Error! 77
Bookmark not defined. but

Author Response

Dear reviewer,

Thank you very much for reviewing our manuscript and for your compliments and suggestions on our work.

  • Incorrect percentages were corrected in table 4. All percentages in both text and tables were checked for any mistakes.
  • The median follow-up is different in the text compared to table 4, because in the text the follow-up of all studies reporting on MIS was mentioned and in table 4 only the studies reporting on recurrence/death rate. To clearify this (and to align the text with table 4), the text is corrected into: "In 10 studies, involving 534 patients, a minimal invasive approach for ART was used. Recurrence and death rates were reported in 9 studies, with a median follow-up of 44 months (range 10-98 months)."
  • "Life birth" was replaced by "live birth" throughout the text and in the supplementary tables.  
  • In line 77, I don't see a formatting issue (anymore), so maybe it is corrected already.  

Reviewer 2 Report

I have read with great interest the long and complex submitted manuscript; the work done is excellent, in which it is possible to underline a conspicuous scientific competence on the topic and a lot of statistical familiarity.
I can only congratulate the authors for what they have produced.
I only ask to review small imperfections in the text, as on page 2 line 77 or some incomplete bibliographic entries, such as item 140, where the year of publication is missing.

Author Response

Dear reviewer,

Thank you very much for reviewing our manuscript and for your compliments on our work.

  • Page 2, line 77: maybe it has already been corrected or the lines have changed, but I don't see an imperfection in the text (anymore)
  • All references were checked and corrected, if incomplete (number 29, 75 and 140)

Reviewer 3 Report

This paper reports on fertility-sparing treatments of ovarian, cervical and endometrial cancers.

The authors have amassed a truly impressive amount of data collected from some 7,699 medical records.

Although a bit redundant at times for the extensive treatment of the topic, the authors’ scholarship and creativity are evident in this manuscript, as evidenced by their methods and reasoning. The data are presented in a clear and orderly fashion.

While an in-depth, systematic review of a single pathology, rather than one of all three categories together, would have been more incisive, the authors have clearly and effectively depicted the current status of fertility-sparing treatments of the three main gynecological cancers.  Their findings define and underscore the critical issues existing in this important topic.

It is my opinion that this paper is suitable of publication in this prestigious journal.

Author Response

Dear reviewer,

Thank you very much for reviewing our manuscript and for your compliments on our work.